# Alkaline Modification of *Arabica*-Coffee and *Theobroma*-Cocoa Agroindustrial Waste for Effective Removal of Pb(II) from Aqueous Solutions

**DOI:** 10.3390/molecules28020683

**Published:** 2023-01-10

**Authors:** Carmencita Lavado-Meza, Leonel De la Cruz-Cerrón, Yvan J.O. Asencios, Francielle Candian Firmino Marcos, Juan Z. Dávalos-Prado

**Affiliations:** 1Escuela Profesional de Ingeniería Ambiental, Universidad Nacional Intercultural de la Selva Central Juan Santos Atahualpa, Chanchamayo 12856, Peru; 2Facultad de Ingeniería, Universidad Continental, Huancayo 12000, Peru; 3Institute of Marine Science, Federal University of São Paulo, Santos 11030-100, SP, Brazil; 4Escola Politecnica, Department of Chemical Engineering, Universidade de São Paulo, Av. Prof. Luciano Gualberto, t. 3, 380, São Paulo 05508-010, SP, Brazil; 5Instituto de Química Física “Rocasolano”, CSIC, 28006 Madrid, Spain

**Keywords:** biosorption, Pb(II) removal, agroindustrial waste, heavy metals

## Abstract

*Arabica*-coffee and *Theobroma*-cocoa agroindustrial wastes were treated with NaOH and characterized to efficiently remove Pb(II) from the aqueous media. The maximum Pb(II) adsorption capacities, q_max_, of *Arabica*-coffee (WCAM) and *Theobroma*-cocoa (WCTM) biosorbents (q_max_ = 303.0 and 223.1 mg·g^−1^, respectively) were almost twice that of the corresponding untreated wastes and were higher than those of other similar agro-industrial biosorbents reported in the literature. Structural, chemical, and morphological characterization were performed by FT-IR, SEM/EDX, and point of zero charge (pH_PZC_) measurements. Both the WCAM and WCTM biosorbents showed typical uneven and rough cracked surfaces including the OH, C=O, COH, and C-O-C functional adsorbing groups. The optimal Pb(II) adsorption, reaching a high removal efficiency %R (>90%), occurred at a pH between 4 and 5 with a biosorbent dose of 2 g·L^−1^. The experimental data for Pb(II) adsorption on WACM and WCTM were well fitted with the Langmuir-isotherm and pseudo-second order kinetic models. These indicated that Pb(II) adsorption is a chemisorption process with the presence of a monolayer mechanism. In addition, the deduced thermodynamic parameters showed the endothermic (Δ*H^0^* > 0), feasible, and spontaneous (Δ*G^0^* < 0) nature of the adsorption processes studied.

## 1. Introduction

Effluents from industrial activities such as smelting, mining, painting, tanning, etc. are causing severe environmental pollution by depositing heavy metals, particularly in aquatic ecosystems [1]. These metals are highly toxic, are not degradable, and can accumulate in living organisms and affect many of their vital functions. Lead (Pb) is the second most toxic metal and can adversely affect the nervous, digestive, and reproductive systems and can even cause death [2,3]. For these reasons and for preventive purposes, for example, the World Health Organization (WHO) has established the permissible limit of Pb in drinking water at 0.01 mg L^−1^ [4].

Various methods are used to remove heavy metals, such as Pb, from wastewater: coagulation–flocculation, liquid–liquid extraction, ion exchange, and electrochemical treatment. However, these methods have disadvantages such as the high operating costs, long operating times, and the generation of a large volume of toxic sludge [5,6]. In this context, the removal of contaminants using biological materials (biosorbents), such as algae, cyanobacteria, fungi, or particularly agroindustrial wastes has become an economical, ecological, and promising alternative method compared with the conventional methods mentioned above [7,8,9,10]. These biosorbents can be modified to improve the adsorption capacity, structural stability, and reusability. The modification can be carried out by chemical reagents (chemical modification), physical calcination, or grinding methods [11]. In particular, Calero et al. [12], Moyo et al. [13], Petrović et al. [14], and Ye and Yu [15], among others, reported that chemical modification, with NaOH, of biosorbent-precursors improved the removal capacity of Pb.

Peru is an important producer of coffee and cocoa worldwide, with annual productions of 136 and 218 Kt, respectively [16]. The processing of coffee cherries and cocoa pods generates wastes at approximately 80% (coffee) [17] and 70% (cocoa) [18,19] of the total weight of the product, constituting a serious environmental problem for their producing regions [20]. However, these agro-industrial residues could be used for the effective cleaning of aqueous ecosystems of the surrounding crops, in which we have evidence that there is contamination with heavy metals such as Pb.

In this work, we significantly improved the absorption capacity of Pb(II) by means of alkaline modification (with NaOH) to coffee- and cocoa-untreated wastes [21].

## *2.* Results and Discussion

### 2.1. Effect of Alkaline Treatment

After treatment with NaOH, both the *arabica*-coffee and *theobroma*-cocoa biosorbents lost weight, 40.2% and 38.4%, respectively (Table 1). This loss may be due to the fact that, during NaOH treatment, the hydrolysis reactions that take place would cause a high dissolution of organic compounds from the biomass and, therefore, its considerable disintegration [12,22].

The basic titrable sites on both WACM- and WTCM-treated biosorbents were almost 1.4 times higher than that in the respective untreated precursors WAC and WTC (see Table 1). According to Santos et al. [22], and Bulgariu and Bulgariu [23], among others, the NaOH treatment provides, due to the hydrolysis reactions, the formation of more carboxylic (-COO^−^) and hydroxyl (-OH) groups (basic titrable sites), both in undissociated as dissociated forms, that improve the Pb-binding properties of WACM and WTCM biosorbents. It is interesting to mention the absence of acid-titrable sites on the surface of each treated biomass (See Table 1).

The point of zero charge, pH_PZC_, values for WACM and WTCM were higher than those for WAC and WTC, respectively (Table 1). This result indicates an increase in the surface basicity of the treated biosorbents, which is consistent with the increase in the concentration of its basic titrable sites, described above. A similar feature was reported by Blázquez et al. [24] for olive stone biomass modified with NaOH.

An interesting consequence of the alkaline treatment of the studied biomasses is the considerable increase in the Pb(II) removal capacity. Thus, taking into account optimal conditions, described below (Section 2.3), the Pb(II) adsorption capacity q_e_ of WACM and WTCM were almost three times higher than of corresponding non-treated WAC and WTC biomasses (see Figure 1). Mangwandi et al. [6], Ren et al. [11], and Gupta et al. [25], among others, reported that chemical modification of a biomass considerably improves its adsorption capacity of heavy metals.

### 2.2. SEM/EDX and FTIR Analysis

SEM micrographs of the WACM and WTCM biosorbents before and after Pb(II) was loaded are shown in Figure 2. Typical uneven and rough surface morphologies are observed. Before adsorption, the images show more porous and less compact structures, than that with Pb(II) loaded. The results of the EDX spectra (Figure 3) confirmed that Pb(II) is adsorbed on the surface of both Pb-loaded biosorbents. Furthermore, the disappearance of the Na peak on these samples (Figure 3 right) can be attributed to ionic exchange with Pb(II). Similar morphologies were reported by Jaihan et al. [5] in papaya peels loaded with Pb(II).

Figure 4 shows the FTIR spectra of WACM and WTCM before and after Pb(II) sorption. The FTIR spectra of the unloaded-Pb samples show the positions of the peaks and absorption bands (in parentheses for WTCM) at the following:

(1) 3339.3 (3335.7) cm^−1^, assignable to typical -OH bond stretching vibrations in samples such as cellulose, lignin, or water [26,27,28,29];

(2) 2919.9 (2910.9) cm^−1^, assignable to the symmetric stretching of the C-H bonds of aliphatic acids [30];

(3) 1636.2 (1621.6) cm^−1^ assignable to the asymmetric stretching of the double bond of C=O carbonyl groups [29];

(4) 1420.4 (1420.4) cm^−1^, assignable to the stretching C-OH and C=O groups of carboxylates [29,31];

(5) 1019.1 (1028.8) cm^−1^, characteristic of C-O-C stretching in polysaccharides [32];

The FTIR spectra after Pb(II) was loaded show changes in the intensity and position of some peaks and bands with respect to those of the clean samples. Thus, for both biosorbents, the positions of the peaks or bands 1, 3, and 4 are displaced with respect to the clean sample values at Δ_1_ = −13.3 (−9.7), Δ_3_ = −4.9 (−23.2), Δ_4_ = −8.5 (−3.6) cm^−1^. These results indicate that the OH, C=O, and C-O groups would be involved in the biosorption of Pb(II). A similar behavior was reported, among others, by Barka et al. [33] and Mahyoob et al. [29] in the removal of Pb(II) by biomasses such as cladodes of prickly pear or olive tree leaves.

### 2.3. Adsorption Experiments

#### 2.3.1. Influence of pH Solution

The pH plays an important role in the adsorption process and provides necessary information on the adsorption–desorption mechanisms. The effect of pH was studied in the range of 2 to 5 (Figure 5) since Pb precipitates, at pH > 5, into Pb(OH)_2_ [27]. The Pb(II) adsorption capacity q_e_, was very low for an acidic medium close to pH 2. It is due to a competing effect between H^+^ and Pb(II) ions for fill surface active sites [30,34]. For pH > 2, q_e_ increased greatly, reaching values of 227.1 and 214.3 mg g^−1^ at pH 4 for WACM and WTCM, respectively. With a further increase at pH 5, the q_e_ values showed improvements by almost 17% for WACM and by only 2% for WTCM. Accordingly, when pH increases, the repulsive interactions between H^+^ and Pb(II) ions decrease, facilitating access of Pb(II) to the surface adsorption sites.

#### 2.3.2. Influence of Biomass Dosage

Figure 6 depicts the Pb(II) removal efficiency, %R, as a function of the biomass dosage. A significant increase in %R is observed for both the WACM and WTCM biosorbents, reaching almost 55% and 60%, respectively, for a biomass dosage of 2 g L^−1^. A further increase in the dosage produces a slight increase in %R for WTCM but a rapid decrease for WACM. The latter trend is due to the agglomeration of the WACM biomass, observed during the experimentation, which would reduce the effective surface area available for the interaction between Pb(II) and the biosorbent [35].

For both the WACM and WTCM biosorbents, the dose of 2 g L^−1^ was selected as the optimal value, which would provide a suitable surface area for the efficient removal of Pb(II).

#### 2.3.3. Influence of Initial Pb(II) ion Concentration, C_0_

The effect of the initial Pb(II) ion concentration C_0_ on the adsorption capacity q_e_ and removal efficiency %R was studied in the range of 5.6 to 130.8 mg L^−1^ (See Figure 7). For both the WACM and WTCM biosorbents, the highest %R values (>92%) were obtained for low C_0_ concentrations (in the range of 5.6 to 40 mg L^−1^), where the ratio of surface active sites to the free Pb(II) ions is high, resulting in rapid adsorption [36]. For high C_0_ concentrations, %R decreased until almost 60% was reached at C_0_ = 130.8 mg L^−1^. The diminishing %R with increasing C_0_ is attributed to the saturation of the available adsorption sites [32,36].

On the other hand, the Pb(II) adsorption capacity q_e_ of both the WACM and WTCM biosorbents showed an opposite trend to %R, since it increases with increasing C_0_. This result can be explained considering that for a given amount of adsorbent, an increase in the amount of Pb(II) ions, in solution, produces a concentration gradient that drives a greater interaction with the active binding sites of the biosorbent [37,38].

### 2.4. Adsorption Isotherms

The adsorption isotherms were studied in a range of initial Pb(II) concentrations C_0_ between 5.6 and 130.8 mg L^−1^, at pH 4 and pH 5, and T = 293 K and t = 120 min. The results are depicted in Figure 8, where Pb(II) adsorption capacity q_e_ vs. C_e_ concentration of Pb(II) in equilibrium is presented. These data were fitted to two very common isotherm models [39,40]: (i) Langmuir model, which assumes solute sorption in monolayers with a homogeneous sorption energy; (ii) Freundlich model, which assumes multilayer sorption, with heterogeneous sorption energies. The adjustment parameters with both models were obtained by fitting the corresponding experimental data [C_e_/q_e_ vs. C_e_] and [log(q_e_) vs. log(C_e_)].

We can see (Table 2) that adsorption isotherms are fitted better with Langmuir (R^2^ close to 1) than the Freundlich model (R^2^ ≤ 0.87). From the first model and, for both pH 4 and pH 5, lower K_L_ and higher maximum sorption capacity q_max_ values are obtained for WACM than for WTCM. At pH 5, q_max_ = 303.0 mg g^−1^ for WACM is almost 21% higher than the value at pH 4, while for WTCM, q_max_ = 223.1 mg g^−1^ is practically the same at both pHs.

q_max_ values of agro-industrial wastes, with alkaline treatment, are consigned in Table 3. We can note that q_max_ of WACM and WTCM are among the highest. On the other hand, it is important to mention that that q_max_ value of the treated biosorbent is almost twice that of its corresponding untreated precursor [21].

### 2.5. Kinetic of Biosorption

The kinetic study is of great importance for the practical and effective use of biosorbents in the industry [41]. In this work, the kinetic studies were carried out by varying the adsorption time from 0 to 180 min, at pH 4; C_0_ = 48.42 and 130.8 mg L^−1^; biosorbent dosage = 2 g L^−1^; and T = 293 K.

The experimental kinetic data were modeled using three adsorption kinetic models (Table 4): pseudo first-order model, pseudo second-order model, and ploting q_t_ vs. t^1/2^ (Weber and Morris model). The parameters obtained after the non-linear adjustments, including correlation coefficient R^2^, are consigned in Table 4. Figure 9 shows the non-linear fit of the pseudo second-order equation to the kinetic data.

For both the WACM and WTCM biosorbents, a better correlation (R^2^ ≈ 1) is obtained with *pseudo*-second-order than the first-order adjustment models. This result indicates that Pb(II) adsorption is a chemisorption process [42]. We can note that the calculated adsorption capacities q_e,cal_ are close to those determined experimentally and, for a given C_0_ concentration, greater for WACM than for WTCM. The adsorption rates (k_2_, rate constant adsorption and h, initial adsorption rate) are comparable for both biosorbents.

The q_t_ vs. t^0.5^ data, for both WACM and WTCM biosorbents, are depicted in Figure 10 and fitted with the intra-particle diffusion Weber–Morris model. According to the k_d_ intra-particle diffusion rate constants (in mg g^−1^ min^−1/2^, Table 4), we can distinguish three parts: The first part shows rapid growth of q_t_ at time t (k_d,I_ > 10.3), particularly at high initial Pb(II) concentrations (C_0_ = 130.8 mg L^−1^), where k_d,I_ can reach values up to seven times higher than for those at low C_0_ concentrations (e.g., 48.4 mg L^−1^). These results would indicate the rapid absorption of Pb(II) ions on the surface of the biosorbents. The second part shows slower growth of q_t_ with t (0.16 < k_d,II_ < 12.4), which would be related to a gradual sorption process, where Pb(II) sorbed would fill the biosorbent pores; this part would be related to the diffusion of Pb(II) inside the biosorbent (intraparticle diffusion) [24]. Finally, the 3rd part shows that q_t_ is practically constant with very low k_d,III_ values (k_d,III_ < 0.6). It indicates that the equilibrium between Pb(II) ions in the solution and the sorbent surface is reached.

### 2.6. Biosorption Thermodynamics

Δ*G*^0^ was calculated from Equations (4) and (5). The plot ln*K*_c_ vs. 1/T (Equation (7)), depicted in Figure 11, was fitted using the least squares method, aiming to calculate the Δ*H*^0^ and Δ*S*^0^ values. The results are consigned in Table 5 and indicate that the Pb(II) adsorption process on both the WACM and WTCM biosorbents is: (i) feasible, spontaneous (positive Δ*G*^0^ values), and more favorable with increasing temperature; (ii) endothermic by nature (positive Δ*H*^0^ values); and (iii) a process with increasing randomness (positive Δ*S*^0^ values) at the solid–liquid interface [43].

Similar results were reported by Song et al. [27], Morosanu et al. [31], Milojkovic et al. [44], among others, for Pb(II) removal by *Auricularia auricular* spent substrate, rapeseed biomass, and *Myriophyllum spicatum* and its compost, respectively. However, a feasible, spontaneous, but exothermic process of Pb(II) adsorption was also reported by Petrović et al. [14] when removing Pb(II) with a hydrochar of grape pomace. Mahyoob et al. [29] determined that Pb(II) adsorption on co-processed olive tree leaves was an exothermic process with decreasing randomness (negative Δ*S^0^*).

## 3. Materials and Methods

### 3.1. Preparation of Biosorbents

*Arabica*-coffee (WAC) and *Theobroma*-cocoa (WTC) waste were obtained from the Satipo and Chanchamayo provinces, respectively, located at Junín, Perú. Both samples were previously washed with abundant distilled water, dried in an oven at 60 °C for 48 h, and finally ground. WAC and WTC were treated with a 0.1 M NaOH solution in a solid–liquid ratio of 1:10 (g biomass: mL solution) for 24 h with constant stirring at 300 rpm. Once treated, both products were filtered and washed with abundant deionized water and then were dried in an oven at 60 °C for 48 h, and finally, both treated samples (WACM and WTCM) were ground again and homogenized with a 70 mesh sieve.

The alkaline treatment of precursor wastes produced a biomass loss. The percentage loss was determined as the difference between the initial sample weight (before treatment) and the final sample weight (after treatment)

All chemical reagents used in this work were of analytical grade.

### 3.2. Biosorbent Characterization

The point of zero charge (pH_PZC_) study was evaluated according to the methodology reported by do-Nascimento et al. [45]. A mixture of 0.05 g of biomass with 50 mL of an aqueous solution under different initial pHs (pH_0_) ranging from 1 to 12 was prepared. The acid dilutions were prepared from a 1 M HCl solution, while basic dilutions were from 1 M NaOH. After 24 h of equilibrium, the final pHs (pH_f_) were measured.The concentrations of the acid and basic groups (or acid/basic titrable sites) on the surface of WACM and WTCM were determined using the Boehm method reported by Aygun et al. [46]. For acid titrable sites (between brackets for basic sites), mixtures of 0.25 g (0.5 g) of the biosorbent with 50 mL of a standardized 0.05 M NaOH (0.1 M HCl) solution were prepared. All the mixtures were shaken, at room temperature, for 24 h at 100 rpm, and then, for each mixture, 20 mL (10 mL) of the supernatant liquid was pipetted and excess acid (base) was adequately titrated using bromocresol blue or phenolphtalien) as an indicator.Fourier transform infrared (FTIR, SHIMADZU IR Affinity) spectroscopy, over a spectral range of 4000 to 500 cm^−1^ was used to characterize the functional groups present on the surface of WACM and WTCM before and after Pb(II) biosorption.Morphological and elemental analysis on the surface of biosorbents were performed by Scanning Electron Microscopy (SEM) coupled with EDX (Energy Dispersive X-rays spectroscopy) (LEO 440 model).

### 3.3. Adsorption Experiments

Batch experiments were carried out using Pb(NO_3_)_2_ solution, with varying Pb(II) concentrations between 5.63 to 130.8 mg L^−1^. The dose of the modified biomass was varied in the range of 0.5 to 6 g L^−1^ and adjusted to a pH in the range of 2.0 to 5.0 (using LAQUA PH1200) by adding 0.01 M HNO_3_ or 0.01 M NaOH. The solutions, at room temperature, were stirred to 150 rpm for 120 min, and the samples were taken at certain time intervals.

The Pb(II) concentrations, before and after adsorption, were evaluated using an Atomic Absorption Spectrophotometer (SHIMADZU-AAS 6800). All adsorption experiments were replicated three times, and the results were averaged.

Pb(II) adsorption capacity q_e_(in mg g^−1^) and removal efficiency (%R) were determined using Equations (1) and (2), respectively [9].
(1)$$q_{e}=\frac{\left(C_{0}-C_{e}\right)\;\times \;V}{m}$$
(2)$$\%R=\frac{{(C}_{0}{-C}_{e})}{C_{0}}\;\times \;100$$
where *C*_0_ and *C*_e_ (in mg·L^−1^) are the initial and equilibrium final Pb(II) concentrations, respectively, and V (in L) is the volume of solution and *m* (in g) is the biosorbent mass.

The adsorption isotherms and kinetic data were obtained by contacting a biomass dose of 2 g L^−1^ with different Pb(II) concentrations at pH 4 (also pH 5 for isotherms). The experimental data were adjusted to the corresponding adsorption models described in Table 6 (isotherms) and Table 7 (kinetic).

### 3.4. Thermodynamic Parameters

This study was carried out by varying the temperature from 293 to 313 K (20 to 40 °C) using a biosorbent dose of 2 g L^−1^, an initial Pb(II) concentration C_0_ = 130.8 mg L^−1^, and a contact time 120 min at pH 4. Parameters such as free energy change Δ*G*°, enthalpy change Δ*H*°, and entropy change, Δ*S*° for the adsorption process studied were calculated using Equations (3)–(6) [47,48]:(3)$$K_{c}=\frac{C_{es}}{C_{e}}$$
(4)$$\Delta G^{o}=-RTLnK_{c}$$
(5)$$\Delta G^{o}=\Delta H^{o}-T\Delta S^{o}$$
(6)$$lnK_{c}=\frac{\Delta S^{o}}{R}-\frac{\Delta H^{o}}{RT}$$
where *R* is the ideal gas constant (8.314 J mol^−1^ K^−1^); *T* is the absolute temperature of the solution; K_c_ is the thermodynamic equilibrium constant; and *C_es_* and *C_e_* are Pb(II) concentrations at equilibrium, respectively, in the biosorbent and solution.

## 4. Conclusions

*Arabica*-coffee (WACM) and *Theobroma*-cocoa (WTCM) biosorbents were chemically modified, with 0.1 M NaOH to improve their Pb(II) adsorption capacities an aqueous medium. After treatment, both WACM and WTCM biosorbents lost weight 40.2% and 38.4%, respectively.

The point of zero charge, pH_PZC_ values at 6 and 6.8, for WACM and WTCM, respectively, were higher than those for the corresponding WAC and WTC untreated samples. These measurements were consistent with the concentration of basic titrable sites, which was almost 1.4 times higher for treated than untreated biosorbents. The basic sites would be associated with the OH, C=O, COH, and C-O-C functional groups, which were identified by FTIR measurements.

SEM/EDX analyses showed typical uneven and rough surface morphologies, more porous and less compact for clean than for Pb(II)-loaded biosorbents.

Both the WACM and WTCM biosorbents reached high Pb(II) removal efficiency %R values (>90%) for a biomass dosage of 2 g L^−1^ at pH between 4 and 5, with initial Pb(II) concentrations C_0_ in the range of 5 to 40 mg L^−1^. For these conditions, the Pb(II) adsorption capacity q_e_ of WACM (and WTCM) was almost three times higher than that of the corresponding untreated biosorbent.

The adsorption isotherm data were well fitted with the Langmuir model (monolayer adsorption mechanism), which provided, at pH 5, maximum Pb(II) adsorption capacities, q_max_, equal to 303.0 and 223.1 mg g^−1^ for WACM and WTCM, respectively. These values are i) twice those corresponding to untreated samples and ii) higher than for other similar alkaline-treated biosorbents reported in the literature.

The adsorption kinetic data were well fitted with the *pseudo*-second-order model, indicating that the Pb(II) adsorption on WACM and WTCM was a chemisorption process. This process, according to our thermodynamic results, can be characterized as endothermic (Δ*H^0^* > 0), feasible, and spontaneous (Δ*G^0^* < 0) and with an increasing randomness (Δ*S^0^* > 0) at the solid–liquid interface.

## Figures and Tables

**Figure 1 molecules-28-00683-f001:**
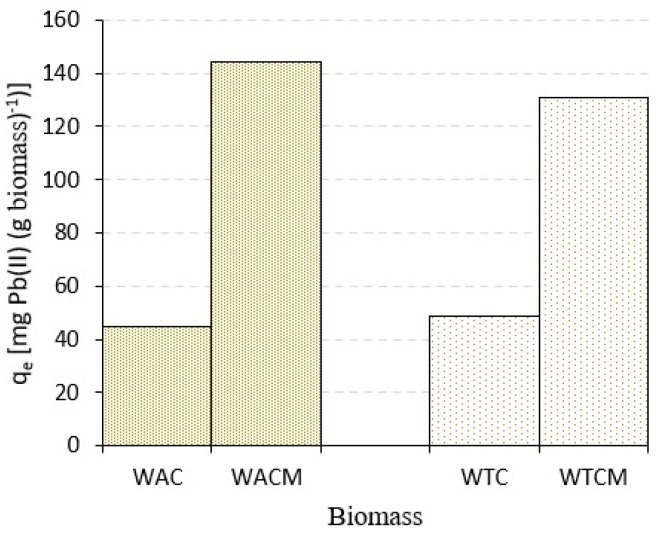
Pb(II) adsorption capacity q_e_ for *Arabica*-coffee and *Theobroma*-cocoa wastes, untreated (WAC and WTC) and treated (WACM and WTCM) biosorbents. Initial Pb(II) concentration, C_0_ = 48.42 mg L^−1^, biosorbent dose = 2 g L^−1^, pH 4.

**Figure 2 molecules-28-00683-f002:**
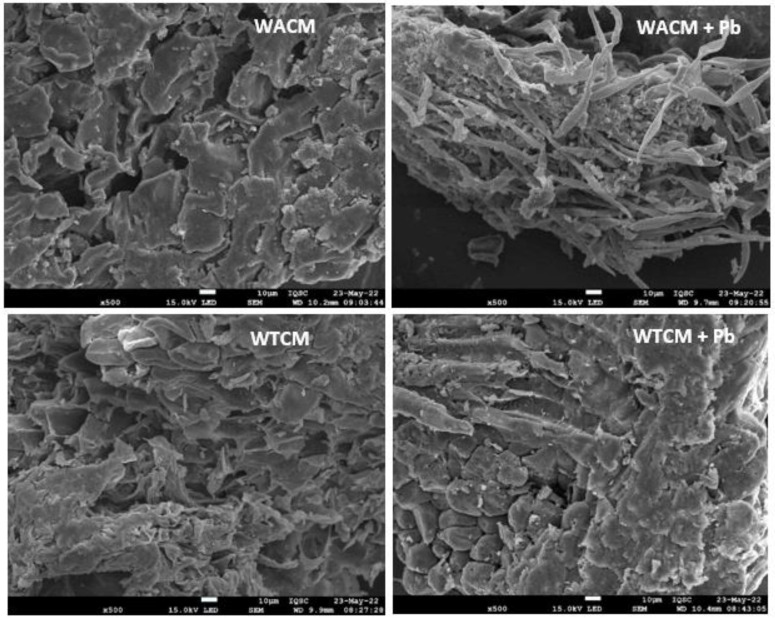
SEM of WACM and WTCM before (**left**) and after (**right**) Pb(II) biosorption. pH = 4, C_0_ = 130.8 mg L^−1^, T = 293 K, t = 120 min.

**Figure 3 molecules-28-00683-f003:**
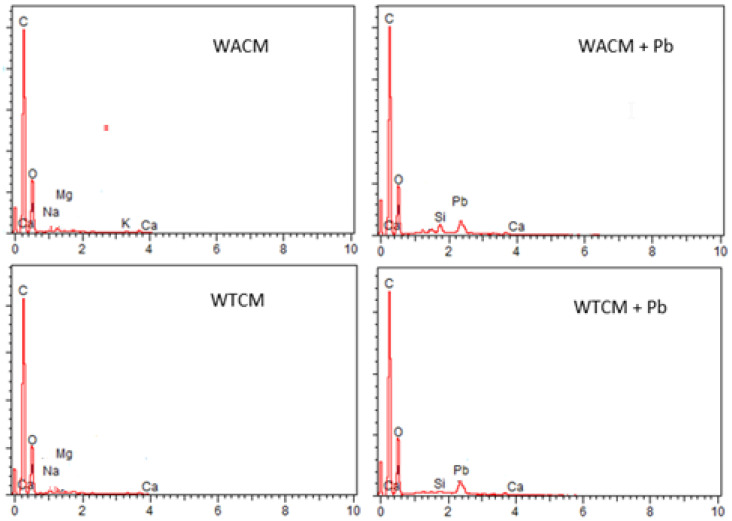
EDX spectra of WACM and WTCM before (**left**) and after Pb adsorption (**right**). pH = 4, C_0_ = 130.8 mg L^−1^, T = 293 K, t = 120 min.

**Figure 4 molecules-28-00683-f004:**
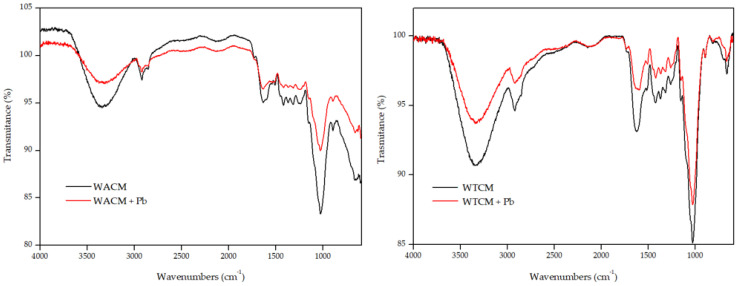
FTIR spectra before (black) and after (red) Pb(II) adsorption by WACM (**left**) and WTCM (**right**). pH 4, C_0_ = 130.8 mg L^−1^, T = 293 K, t = 120 min.

**Figure 5 molecules-28-00683-f005:**
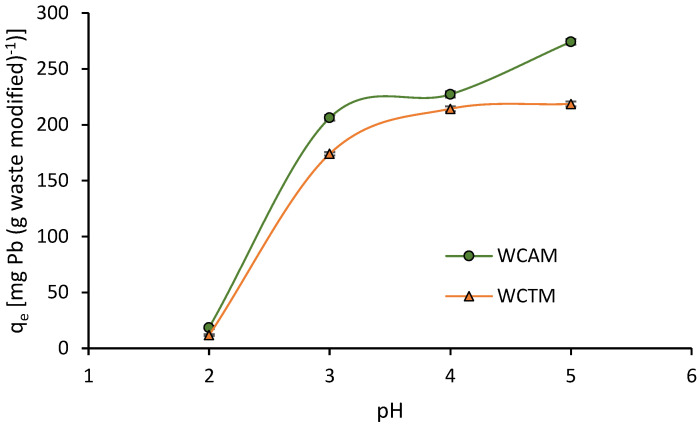
Influence of pH on the Pb(II) adsorption capacity, q_e_, for T = 293 K, adsorption time = 120 min, biosorbent dose = 2 g L^−1^, C_0_ = 130.8 mg L^−1^.

**Figure 6 molecules-28-00683-f006:**
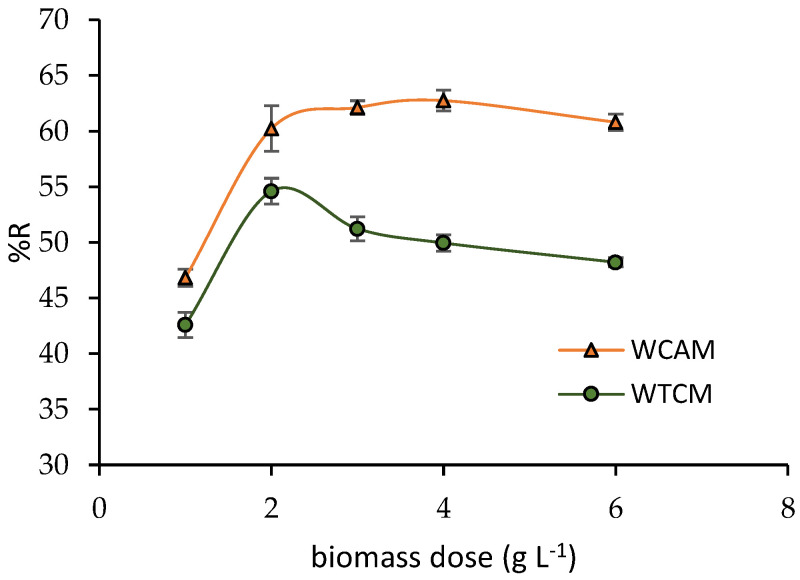
Effect of biomass dosage. T = 293 K, adsorption time = 120 min, pH = 4, C_0_ = 130.8 mg L^−1^.

**Figure 7 molecules-28-00683-f007:**
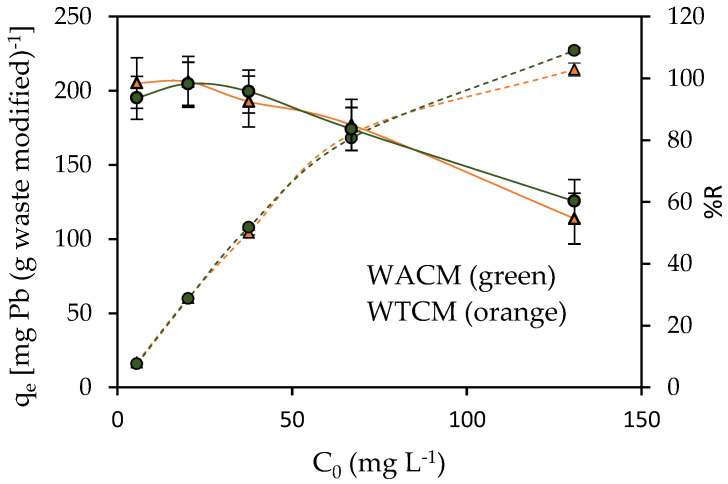
Effect of the initial Pb(II) concentration C_0_ on q_e_ (dotted lines) and %R (full lines). *t* = 60 min, T = 293 K, pH 4, biosorbent dose = 2 g L^−1^.

**Figure 8 molecules-28-00683-f008:**
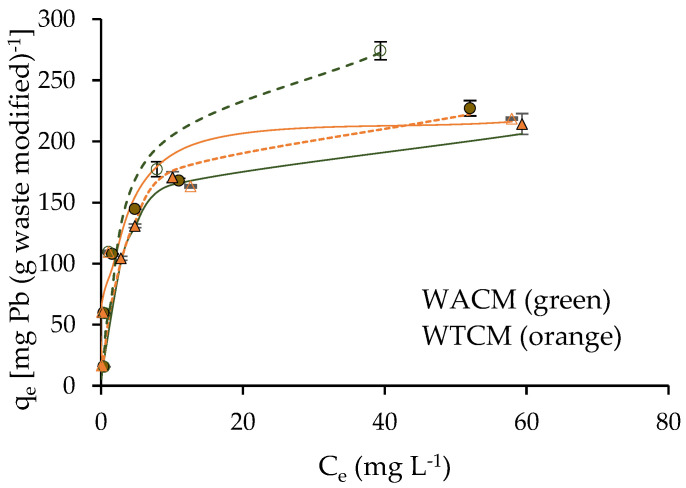
WACM and WTCM adsorption isotherms fitted to Langmuir model. For pH 4 (continuous lines) and pH 5 (dotted lines); biosorbent dose = 2 g L^−1^, t = 120 min, T = 293 K.

**Figure 9 molecules-28-00683-f009:**
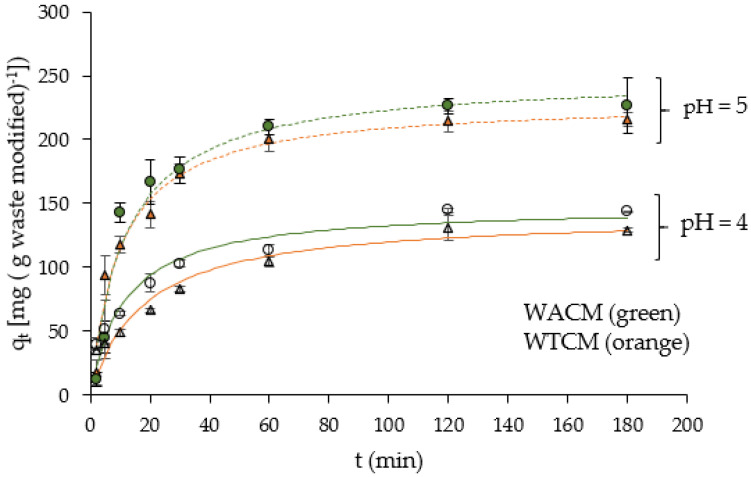
q_t_ vs. time t. C_0_ = 48.4 (continuous lines) and 130.8 mg L^−1^ (dotted lines); dose = 2 g L^−1^, T = 293 K. q_t_ = amount Pb(II) removed per mass unit of biosorbent at time t.

**Figure 10 molecules-28-00683-f010:**
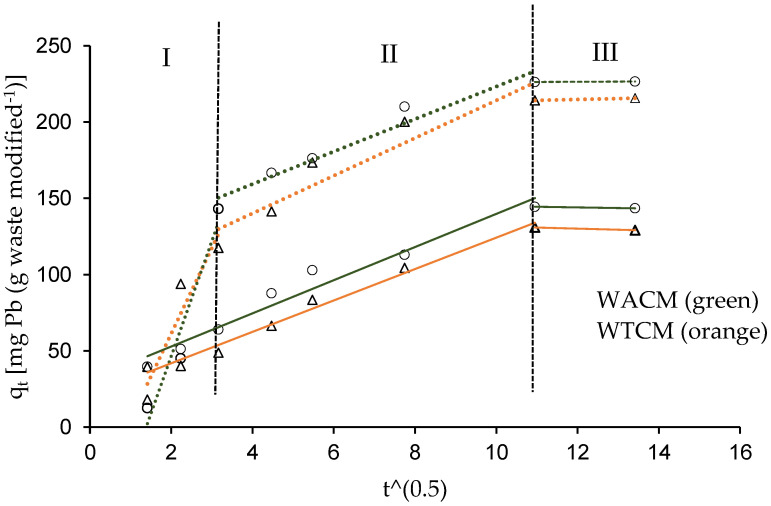
Weber–Morris plots of Pb(II) adsorption on WACM and WTCM. C_0_ = 48.4 (continuous lines) and 130.8 mg L^−1^ (dotted lines).

**Figure 11 molecules-28-00683-f011:**
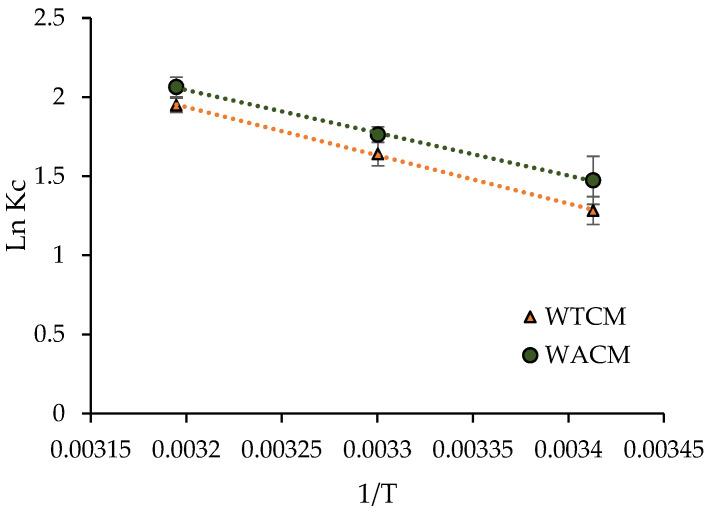
lnK_c_ vs. 1/T plot for Pb(II) biosorption onto WTCM and WACM at pH = 4, biosorbent dosages = 2 g L^−1^, t = 120 min, C_0_ = 130.8 mg L^−1^.

**Table 1 molecules-28-00683-t001:** Physical–chemical characteristics of untreated and alkaline treated *Arabica*-coffee and *Theobroma*-cocoa wastes.

	WAC Coffe Waste	WTC Cocoa Waste	WACM Coffe Waste	WTCM Cocoa Waste
	Untreated		Alkaline Treated ^a^	
Point of Zero Charge, pH_PZC_	4.8	6	6	6.8
Acid titrable sites (mmol g^−1^)	2.8 × 10^−2^	1.97 × 10^−2^	0	0
Basic titrable sites (mmol g^−1^)	2.12 × 10^−2^	1.84 × 10^−2^	2.97 × 10^−2^	2.63 × 10^−2^
% Biomass loss due to treatment			40.2	38.4

^a^ 0.1 M NaOH.

**Table 2 molecules-28-00683-t002:** Isothermal parameters for Pb(II) adsorption on WACM and WTCM, adjustment to Langmuir and Freundlich models.

		WACM		WTCM	
	Parameters	pH = 4	pH = 5	pH = 4	pH = 5
Langmuir model	K_L_ (L·mg^−1^) q_max_ (mg·g^−1^) R^2^	0.32 238.1 ~1	0.22 303.0 0.98	0.45 222.2 ~1	0.61 223.1 ~1
Freundlich model	K_F_ (mg·g^−1^ L^(1/n)^·mg^−(1/n)^) n_F_ R^2^	58.47 2.39 0.73	56.78 2.04 0.72	66.08 2.80 0.87	71.69 3.11 0.79

**Table 3 molecules-28-00683-t003:** Comparative table of the maximum adsorption capacity q_max_ of Pb(II), for biosorbents with alkaline treatment.

Biosorbent Wastes	q_max_ (mg g^−1^)	Reference
Apricot shells Mangifera indica seed shells Olive tree pruning Grape pomace Moringa oleifera tree leaves *Theobroma* cacao; WTCM *Arabica* coffee; WACM	37.37 59.25 121.60 137 209.54 223.1 303.0	[35] [13] [12] [14] [36] This work This work

**Table 4 molecules-28-00683-t004:** Kinetic parameters of Pb(II) adsorption on WACM and WTCM biosorbents.

Model	Parameters	WACM		WTCM	
		C_0_ = 130.8 mg L^−1^	C_0_ = 48.42 mg L^−1^	C_0_ = 130.8 mg L^−1^	C_0_ = 48.42 mg L^−1^
Pseudo 1st order	k_1_(min^−1^) q_e,cal_ (mg g^−1^) ^a^ R^2^	0.075 206.62 0.94	0.063 132.93 0.84	0.075 206.62 0.94	0.043 124.92 0.87
Pseudo 2nd order	k_2_ (g mg^−1^ min^−1^) q_e,cal_ (mg g^−1^) ^a^ h R^2^	0.0003 249.53 20.99 0.94	0.0006 147.98 13.14 0.93	0.0004 229.99 21.16 0.97	0.0004 141.23 7.98 0.92
Weber and Morris model	k_d,I_ (mg g^−1^ min^−1/2^) ^b^ R^2^ k_d,II_ (mg g^−1^ min^−1/2^) ^b^ R^2^ k_d,III_ (mg g^−1^ min^−1/2^) ^b^ R^2^	75.4 0.94 10.7 0.95 0.12 1	10.9 0.96 0.4 1 - -	56.2 0.90 12.4 0.89 0.58 1	10.3 0.99 0.7 1 - -

^a^ Calculated adsorption capacity. ^b^ Intraparticle diffusion rate constant.

**Table 5 molecules-28-00683-t005:** Thermodynamic parameters for Pb(II) adsorption on WACM and WTCM.

	∆*H*^0^ (kJ mol^−1^)	∆*S*^0^ (J mol^−1^ K^−1^)	∆*G*^0^ (kJ mol^−1^)		
			293 K	303 K	313 K
WACM	22.5	88.9	−35.90	−44.42	−53.70
WTCM	25.4	97.2	−31.26	−41.32	−50.70

**Table 6 molecules-28-00683-t006:** Adsorption isotherm models.

Model	Equation	Parameters
Langmuir	$\frac{C_{e}}{q_{e}}=\frac{1}{q_{máx}k_{L}}+\frac{C_{e}}{q_{máx}}$	q_e_ (mg g^−1^): adsorption capacity C_e_ (mg L^−1^): adsorbate concentration in equilibrium q_max_ (mg g^−1^): maximum sorption capacity k_L_ (L mg^−1^): Langmuir constant related to the affinity between sorbent and sorbate
Freundlich	$\ln q_{e}=\ln k_{F}+\frac{1}{n}\ln C_{e}$	k_F_ (mg g^−1^ L^(1/n)^·mg^−(1/n)^): equilibrium constant n: constant related to the affinity between sorbent and sorbate

**Table 7 molecules-28-00683-t007:** Kinetic adsorption models.

Model	Equation	Parameters
Pseudo-first order	$q_{t}=q_{e}\left(1-e^{-K_{1}t}\right)$	q_e_ (mg g^−1^): adsorption capacity q_t_ (mg g^−1^): amount of Pb(II) retained per unit biomass at time t. k_1_ (min^−1^): first-order kinetic constant k_2_ (g (mg min)^−1^): rate constant adsorption h (mg(g min)^−1^): initial adsorption rate
Pseudo-second order	$q_{t}=q_{e}\frac{q_{e}K_{2}t}{1+q_{e}K_{2}t}\;h=k_{2}q_{e}^{2}$	
Weber and Morris	$q_{t}=k_{d}t^{1/2}+B$	k_d_ (mg g^−1^ min^−1/2^): intraparticle diffusion rate constant B (mg g^−1^): constant related to the thickness of the adsorbent boundary layer

## Data Availability

Not applicable.

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
