# Peer review of "Alkaline Modification of Arabica-Coffee and Theobroma-Cocoa Agroindustrial Waste for Effective Removal of Pb(II) from Aqueous Solutions"

_molecules, 2023, doi:10.3390/molecules28020683_

Round 1

Reviewer 1 Report

1- the authors failed to include the materials and methods section in the manuscript, they are requested to include it.

2- Authors must correctly number tables throughout the manuscript. For example, on line 73, page 2, they refer to table 3, but this table does not correspond to the information they mention, in addition to table 1, they jumped to table 3 and omitted table 2, that happens later too, so they should check this error throughout the manuscript.

3- Instead of "topography" it should be morphology, correct on the line99 page 3.

4- I do not agree with the affirmation that the authors make in line 103 page 3, I consider that the presence of sodium is due to the fact that the material after being treated with NaOH perhaps did not wash enough and later in the medium I accused this sodium " it came out" of the structure, but I don't think it is related to an ion exchange.

5-What is the relevant conclusion that can be drawn from the XRD results? Because as far as I read they only use it to mention the presence of the characteristic peaks of biomaterials and not even a lead face appears. Consider that does not provide relevant information for this analysis

6- The structure of the manuscript is disordered and is not typical of topics on adsorption, first the characterization of the material is established, then the studies of the effect of variables that influence the adsorption process, such as pH, are mentioned, and followed by the kinetics. adsorption in order to know the equilibrium time and is used to apply mathematical models and analyze the system. Subsequently, the results of the adsorption isotherm and the application and discussion of the results when applying the mathematical models to the experimental data are presented. I suggest that the authors respect this order and modify the structure of the article that they reflect. I reiterate that it is necessary to include the section on materials and methods.

7- On line 146 page 6 the authors mention that Pb can precipitate in the Pb(OH)+ form, however this cannot be a precipitated form because it is in the ion form. Check and correct the respect.

8- If the highest adsorption is achieved at pH=%, why do the authors report that they performed adsorption at pH=4 in Figure 1?

9-Why do the authors carry out the pH, FTIR, material dose and initial concentration studies using different initial concentrations? This is a mistake because they should always use the same initial concentration value, in addition, the isotherm studies are done much higher concentration range. Could you explain this and correct it?

10- The authors mention, with respect to the analysis of the results that show in figure 8 that the %R decreases when the adsorption capacity for that point increased?, with which I do not agree and it seems incongruous to me, could they explain this statement and the reasoning to understand it?

11- In the figure 9 the authors must identify which graph corresponds to pH=4 and which to pH=5.

11-On line 203 page 8, the authors state "these KL and qmax values indicate high affinity between Pb(II) sorbate and the biosorbent studied", however qmax gives different information on the affinity of the adsrobate for the adsorbent. The authors must correct this error or write in another way so that they do not misinterpret what they want to express in this regard.

12- Why do the authors in figure 1 report an initial concentration of 48.42 mg/L and in line 216 page 9 report 48.4 mg/L?

13-In figure 10 it is not clear to me what models were applied, whether linear or non-linear, could you include all the information that is missing in the document?

14- Why do the authors decide to do the kinetics for two initial concentrations and the isotherms for two pH values? Everything must be done under the same conditions because otherwise the correct comparisons and analyzes cannot be made. Furthermore, if the authors had already studied the influence of the initial concentration, why repeat the experiments, now as kinetics?

15- Why do the authors show 4 curves or lines in figure 11 and in table 4 report the values of the variables of the intraparticle diffusion model for only three? in the text they mention the model of Weber and Morris and in the table it has another name, so what is it???, where do the values of k2 and h that the authors mention on line 232 page 10 come from?

16- On line 246 page 10 the authors mention "This part would be related to the diffusion of Pb(II) inside the biosorbent (intraparticle diffusion)", however, I disagree because the last zone is reported to be characteristic of the adsorption process , but not diffusion as they indicate. I believe that they should review the literature in this regard and not make misconceptions

17- The materials and methods section should go before the results, which is why it mentioned before that it did not appear in the document. The authors must respect the order that a manuscript has.

18- Instead of writing "hours" the correct form is "h". This error must be corrected throughout the manuscript.

19- In the methodology that they describe about the pH at the point of zero charge, the authors omit the reagent that maintains the constant ionic strength. They must indicate which one it was, because it would not be correct if they omitted this part in the experiment.

20- Why do the authors use for a 0.05 M NaOH solution, a 0.1 M HCl solution and then in line 294 page 12 they report that the HCl concentration is 0.05?

21- Although the document is true, it shows two residues that can be revalued, which I consider to be the novelty of the article. In the development and discussion of the same, there is a lot of information and there are errors of concept and order in the document. I believe that the document should be exhaustively reviewed and redrafted, because as it is presented it is not suitable to be considered for publication.

Author Response

Point 1: the authors failed to include the materials and methods section in the manuscript, they are requested to include it..

Response 1: It has been corrected and included in item 3.

Point 2: Authors must correctly number tables throughout the manuscript. For example, on line 73, page 2, they refer to table 3, but this table does not correspond to the information they mention, in addition to table 1, they jumped to table 3 and omitted table 2, that happens later too, so they should check this error throughout the manuscript.

Response 2:They have been corrected

Point 3: Instead of "topography" it should be morphology, correct on the line99 page 3.

Response 3: It has been corrected

Point 4: - I do not agree with the affirmation that the authors make in line 103 page 3, I consider that the presence of sodium is due to the fact that the material after being treated with NaOH perhaps did not wash enough and later in the medium I accused this sodium " it came out" of the structure, but I don't think it is related to an ion exchange.

Response 4:  We have checked and tested the Na presence. The material was exhaustively washed, in addition other authors (mentined in this paper) also reported the presence of this metal on different biosorbents.

Point 5: What is the relevant conclusion that can be drawn from the XRD results? Because as far as I read they only use it to mention the presence of the characteristic peaks of biomaterials and not even a lead face appears. Consider that does not provide relevant information for this analysis.

Response 5: We appreciate the referee's observation. The XRD results have been removed

Point 6: The structure of the manuscript is disordered and is not typical of topics on adsorption, first the characterization of the material is established, then the studies of the effect of variables that influence the adsorption process, such as pH, are mentioned, and followed by the kinetics. adsorption in order to know the equilibrium time and is used to apply mathematical models and analyze the system. Subsequently, the results of the adsorption isotherm and the application and discussion of the results when applying the mathematical models to the experimental data are presented. I suggest that the authors respect this order and modify the structure of the article that they reflect. I reiterate that it is necessary to include the section on materials and methods.

Response 6: All these observations have been taken into account.

Point 7: On line 146 page 6 the authors mention that Pb can precipitate in the Pb(OH)+ form, however this cannot be a precipitated form because it is in the ion form. Check and correct the respect

Response 7: It has been corrected

.

Point 8: If the highest adsorption is achieved at pH=%, why do the authors report that they performed adsorption at pH=4 in Figure 1?

Response 8: We have found that the optimal Pb(II) adsorptions (%R > 90%) are reached at pH between 4 and 5. We elected pH 4 as a typical comprartive picture of Pb(II) adsorption capacity qe for treated and untreated biosorbents.

Point 9: Why do the authors carry out the pH, FTIR, material dose and initial concentration studies using different initial concentrations? This is a mistake because they should always use the same initial concentration value, in addition, the isotherm studies are done much higher concentration range. Could you explain this and correct it?

Response 9: We appreciate the referee's observation and in this context, we have taken into account for FTIR analysis; pH and dose effects, the same inicitial concentration which one has been obtained high Pb(II) adsorption capacity values. For isotherm studies and in order to obtain wide range de equilibrium concentrations, Ce, have been considered initial concentrations range from 5.6 to 130.8 mg L-1.

Point 10: The authors mention, with respect to the analysis of the results that show in figure 8 that the %R decreases when the adsorption capacity for that point increased?, with which I do not agree and it seems incongruous to me, could they explain this statement and the reasoning to understand it?

Response 10: It is important to note that the removal efficiency %R depends only on the Pb(II) initial and final concentrations; while the adsorption capacity qe also includes the parameter of biosorbent dose. Therefore, there is not necessarily a direct relationship between %R and qe

Point 11: On line 203 page 8, the authors state "these KL and qmax values indicate high affinity between Pb(II) sorbate and the biosorbent studied", however qmax gives different information on the affinity of the adsrobate for the adsorbent. The authors must correct this error or write in another way so that they do not misinterpret what they want to express in this regard

Response 11: It has been corrected.

Point 12: Why do the authors in figure 1 report an initial concentration of 48.42 mg/L and in line 216 page 9 report 48.4 mg/L?

Response 12: It has been corrected

Point 13: -In figure 10 it is not clear to me what models were applied, whether linear or non-linear, could you include all the information that is missing in the document?

Response 13: It has been corrected

Point 14: Why do the authors decide to do the kinetics for two initial concentrations and the isotherms for two pH values? Everything must be done under the same conditions because otherwise the correct comparisons and analyzes cannot be made. Furthermore, if the authors had already studied the influence of the initial concentration, why repeat the experiments, now as kinetics?

Response 14: We have evaluated the biosorption kinetics at, lowest and highest, initial concentrations in order to confirm that he same kinetic behavior is observed for both conditions. On the other hand, we have evaluated the isotherms at pH 4 and 5 values, because the greater biosorption capacities are obtained at these pH values.

Point 15: Why do the authors show 4 curves or lines in figure 11 and in table 4 report the values of the variables of the intraparticle diffusion model for only three? in the text they mention the model of Weber and Morris and in the table it has another name, so what is it???, where do the values of k2 and h that the authors mention on line 232 page 10 come from?

Response 15: We have defined three regions or stages to study, at two different Pb(II) initial concentrations, the absorption kinetics by means Intra-particle diffusion Weber –Morris model. Therefore each region is characterized by the corresponding k value.

Point 16: On line 246 page 10 the authors mention "This part would be related to the diffusion of Pb(II) inside the biosorbent (intraparticle diffusion)", however, I disagree because the last zone is reported to be characteristic of the adsorption process , but not diffusion as they indicate. I believe that they should review the literature in this regard and not make misconceptions.

Response 16: It has been corrected

Point 17: The materials and methods section should go before the results, which is why it mentioned before that it did not appear in the document. The authors must respect the order that a manuscript has.

Response 17: It has been corrected. The materials and methods section is considered in item 3.

Point 18: Instead of writing "hours" the correct form is "h". This error must be corrected throughout the manuscript.

Response 18: It has been corrected

Point 19: In the methodology that they describe about the pH at the point of zero charge, the authors omit the reagent that maintains the constant ionic strength. They must indicate which one it was, because it would not be correct if they omitted this part in the experiment.

Response 19: The methodology reported by Nascimento was used, in which the reagent stated by the reviewer is not mentioned.

Point 20: Why do the authors use for a 0.05 M NaOH solution, a 0.1 M HCl solution and then in line 294 page 12 they report that the HCl concentration is 0.05?

Response 20: It has been corrected

Reviewer 2 Report

The article deals with biosorption of lead from aqueous solution by sorbent based on alkaline modified arabica-coffee and theobroma-cocoa agroindustrial waste. The manuscript addresses an issue that is very important and relevant to wastewater treatment. Results in this manuscript are worthy; still some corrections should be made. My suggestion is to accept manuscript with major revision.

Title: Alkaline modification of arabica-coffee and theobroma-cocoa agroindustrial waste for effective removal of Pb(II) from aqueous solutions

Article Type: Original paper

Introduction

Line 47: use more appropriate word instead biological matrices (e.g various biological materials).

Line 57: introduce reference for amount of annually produced coffee and cocoa in Peru.

Results and Discussion

Line 80: pHpzc - the point of zero charge

Line 114: Authors should have examined and compared effect of alkaline treatment on crystalline structure of raw materials, not alkali treated materials before and after lead sorption. In first case they would have noticed differences in their crystal structure: loss of amorphous phase and increase of crystallinity. This paper describes and compares similar materials and treatment: Šoštarić Tatjana, Petrović Marija, Stojanović Jovica, Marković Marija, Avdalović Jelena, Hosseini-Bandegharaei Ahmad, Lopičić Zorica, Structural changes of waste biomass induced by alkaline treatment: the effect on crystallinity and thermal properties, Biomass Conversion and Biorefinery (2022) 12:2377-2387; https://doi.org/10.1007/s13399-020-00766-2

Line 143: In order to examine the sorption process it is necessary to determine the effects of different operating parameters such as pH, sorbent concentration, metal concentration and contact time. The batch adsorptions studies identify the optimum parameters affecting the efficiency of metal removal by adsorbent. All those experiments should be done within same conditions, varying just one of parameters. In this paper authors exam effect of pH on biosorption capacity at initial lead concentration of 67 mg/L (line 156), biomase dosage at initial lead concentration of 50 mg/L (line 168), while effect of contact time at initial lead concentration of 48.4 mg/L and 130.8 mg/L (line 224).

My suggestion is to take a look on some typical biosorption papers from authors references list and to organise manuscript according to them: e.g. after characterisation (line 143) introduce subsection “Adsorption experiments” (like it is in Materials and Methods) and describe results of effect of solution pH, biosorbent dosage, contact time and initial concentration (suggestion is to show Figure 6,7,8 and 10 in one Figure) on biosorption, and then according to obtained results (when optimal parameters are determined) exam and present results of fitting by isotherm and kinetic models.   

Line 185: In Figure 9 authors should present both isotherm curves (Langmuir and Freundlich)

Line 211: Table 3: reference 34: alkali modified apricot shells have maximum sorption capacity: qmax = 37.37 mg/g.

Line 226: Author should have used non-linear forms of kinetic models, because many researchers agree that the non-linear forms of PFO and PSO models are more suitable than the linear forms for fitting the experimental data. This paper deals with that topic: Lin, J., Wang, L. Comparison between linear and non-linear forms of pseudo-first-order and pseudo-second-order adsorption kinetic models for the removal of methylene blue by activated carbon. Front. Environ. Sci. Eng. China 3, 320–324 (2009). https://doi.org/10.1007/s11783-009-0030-7. Also, author should check the results for adsorption capacities obtained from PFO kinetic model because they are too high (from 12 to 40 g/g). Please check and correct.

Materials and Methods

Line 272: provide information of equipment (grounder, pH meter).

Line 280 equation 1 is not necessary

Line 283: pHpzc - the point of zero charge

Line 299: Since the Energy Dispersive X-Ray Analysis (EDX), is an x-ray technique used to identify the elemental composition of materials, please change elemental analysis in text by elemental composition.

Line 305: Change sentence “The biosorption data from aqueous solutions were obtained using the immersion method” with some more appropriate (e.g. the biosorption of lead onto treated biomasses was investigated by a batch system…or something similar).

Line 306: sorbent dosage is from 0.5 g/L to 6 g/L.

Author Response

Point 1: use more appropriate word instead biological matrices (e.g various biological materials)

Response 1: It has been corrected

Point 2: introduce reference for amount of annually produced coffee and cocoa in Peru.

Response 2: It has been corrected

Point 3: Instead of "topography" it should be morphology, correct on the line99 page 3.

Response 3: It has been corrected

Point 4: Line 80: pHpzc - the point of zero charge.

Response 4: It has been corrected

Point 5: Line 114: Authors should have examined and compared effect of alkaline treatment on crystalline structure of raw materials, not alkali treated materials before and after lead sorption. In first case they would have noticed differences in their crystal structure: loss of amorphous phase and increase of crystallinity. This paper describes and compares similar materials and treatment: Šoštarić Tatjana, Petrović Marija, Stojanović Jovica, Marković Marija, Avdalović Jelena, Hosseini-Bandegharaei Ahmad, Lopičić Zorica, Structural changes of waste biomass induced by alkaline treatment: the effect on crystallinity and thermal properties, Biomass Conversion and Biorefinery (2022) 12:2377-2387; https://doi.org/10.1007/s13399-020-00766-2

Response 5:  According with Reviewer #1, we have considered that XRD analysis does not provide relevant information. Thefore, we have removed this part.

Point 6: Line 143: In order to examine the sorption process it is necessary to determine the effects of different operating parameters such as pH, sorbent concentration, metal concentration and contact time. The batch adsorptions studies identify the optimum parameters affecting the efficiency of metal removal by adsorbent. All those experiments should be done within same conditions, varying just one of parameters. In this paper authors exam effect of pH on biosorption capacity at initial lead concentration of 67 mg/L (line 156), biomase dosage at initial lead concentration of 50 mg/L (line 168), while effect of contact time at initial lead concentration of 48.4 mg/L and 130.8 mg/L (line 224).

My suggestion is to take a look on some typical biosorption papers from authors references list and to organise manuscript according to them: e.g. after characterisation (line 143) introduce subsection “Adsorption experiments” (like it is in Materials and Methods) and describe results of effect of solution pH, biosorbent dosage, contact time and initial concentration (suggestion is to show Figure 6,7,8 and 10 in one Figure) on biosorption, and then according to obtained results (when optimal parameters are determined) exam and present results of fitting by isotherm and kinetic models. 

 .

Response 6: We appreciate the referee's observation and in this context, we have taken into account for pH or dose effects, the same inicitial concentrations which one has been obtained high Pb(II) adsorption capacity values. We have evaluated the biosorption kinetics at, lowest and highest, initial concentrations in order to confirm that he same kinetic behavior is observed for both conditions. On the other hand, we have evaluated the isotherms at pH 4 and 5 values, because the greater biosorption capacities are obtained at these pH values

Point 7: Line 185: In Figure 9 authors should present both isotherm curves (Langmuir and Freundlich)

Response 7: The data are better adjusted to the Langmuir isotherm, which is why only this isotherm has been depicted. We believe that representing both models could confuse the reader.

.

Point 8: Line 211: Table 3: reference 34: alkali modified apricot shells have maximum sorption capacity: qmax = 37.37 mg/g

Response 8: It has been corrected

.

Point 9: Line 226: Author should have used non-linear forms of kinetic models, because many researchers agree that the non-linear forms of PFO and PSO models are more suitable than the linear forms for fitting the experimental data. This paper deals with that topic: Lin, J., Wang, L. Comparison between linear and non-linear forms of pseudo-first-order and pseudo-second-order adsorption kinetic models for the removal of methylene blue by activated carbon. Front. Environ. Sci. Eng. China 3, 320–324 (2009). https://doi.org/10.1007/s11783-009-0030-7. Also, author should check the results for adsorption capacities obtained from PFO kinetic model because they are too high (from 12 to 40 g/g). Please check and correct.

Response 9: It has been corrected

Round 2

Reviewer 1 Report

The observation were taking into account. he manuscript is acceptable for publication.

Reviewer 2 Report

Please remove sentence from line 374: "XRD pattern for all 374 samples showed typical patterns of amorphous structures, which would include lignocellulosic compounds". 

line 227: correct "weber and Morris model" into Weber and Morris model